# Regulatory gene function handoff allows essential gene loss in mosquitoes

Alys M. Cheatle Jarvela[1], Catherine S. Trelstad[1] & Leslie Pick [1✉]

Regulatory genes are often multifunctional and constrained, which results in evolutionary conservation. It is difficult to understand how a regulatory gene could be lost from one species' genome when it is essential for viability in closely related species. The gene *paired* is a classic *Drosophila* pair-rule gene, required for formation of alternate body segments in diverse insect species. Surprisingly, *paired* was lost in mosquitoes without disrupting body patterning. Here, we demonstrate that a *paired* family member, *gooseberry*, has acquired *paired*-like expression in the malaria mosquito *Anopheles stephensi*. *Anopheles-gooseberry* CRISPR-Cas9 knock-out mutants display pair-rule phenotypes and alteration of target gene expression similar to what is seen in *Drosophila* and beetle *paired* mutants. Thus, *paired* was functionally replaced by the related gene, *gooseberry*, in mosquitoes. Our findings document a rare example of a functional replacement of an essential regulatory gene and provide a mechanistic explanation of how such loss can occur.

[1] Department of Entomology, University of Maryland, Collage Park, MD, USA. ✉email: lpick@umd.edu

Gene duplication followed by divergence has been explored as a mechanism for increasing genomic complexity and promoting adaptation. In 1970, Ohno posed the revolutionary idea that after a gene is duplicated creating paralogs, although frequently one redundant copy is rapidly lost, sometimes this new gene copy becomes the raw material for evolution of a new structure or function[1]. Later, Force et al. expanded upon this idea, postulating that the two redundant copies could divide ancestral functions amongst themselves, especially through modification of their expression domains, which relaxes the evolutionary constraint on both paralogs[2]. These two mechanisms are now referred to as neofunctionalization and subfunctionalization, respectively.

*Drosophila melanogaster* (*Dmel*) segmentation genes *paired* (*prd*) and *gooseberry* (*gsb*), encoding paralogous Pax3/7 family transcription factors, were among the first genes demonstrated to have diverged by subfunctionalization[3]. Both genes were discovered in a genetic screen in *Drosophila* for embryonic lethal mutations that disrupt body patterning. However, their roles in segmentation are distinct: *Dmel-prd* is a classic pair-rule gene, with mutants lacking alternate segment primordia. In contrast, *Dmel-gsb* is a segment polarity gene, with mutants displaying identical defects in each body segment. These genes were later shown to have distinct developmental expression patterns, with *prd* expressed in a seven-stripe pattern during blastoderm stage in the primordia of body regions missing in *prd* mutants[4]. After this, *gsb* is expressed in a 14-stripe pattern, emerging during germband extension[5]. Experiments in the Noll lab demonstrated that these spatiotemporal differences in expression pattern, as opposed to unique protein sequences, account for Prd and Gsb's distinct roles in the segmentation process in *Drosophila*[3].

More recently, segmentation mechanisms have been explored in a variety of other insects, and many also show signs of *prd* and *gsb* subfunctionalization, suggesting that this phenomena is not specific to *Drosophila* or even flies in general. The function of *prd* as a pair-rule gene appears to be ancestral within Holometabola, the clade of insects that metamorphose from larval to adult forms. *Dmel*-Prd's pair-rule function is shared by beetles, the only other member of this clade in which functional tests have been carried out. Notably, these results apply to two long-diverged beetle species, the flour beetle, *Tribolium castaneum* (*Tcas*), and the hide beetle *Dermestes maculatus* (*Dmac*). *Tcas-prd* mutant and *Tcas*- and *Dmac-prd* RNAi knockdown beetle embryos all die before hatching due to loss of alternate segments[6–9]. Preliminary evidence shows that while *Tcas-gsb* knockdown is lethal and produces an abnormally segmented embryo, the phenotype is not pair-rule[10]. In addition, Prd's function in promoting segment formation in flies as well as beetles is mediated partly through positive regulation of downstream target genes, such as the segment polarity gene, *engrailed* (*en*)[8,9,11]. Half of the *en* stripes fail to develop in *prd*-deficient fly or beetle embryos. In sum, *prd* mutant flies and both mutant and RNAi knockdown beetle embryos die before hatching due to conserved pair-rule defects with loss of alternate segments and *en* stripes[7–9,11,12]. Moreover, there is evidence of subfunctionalization at the gene expression level in a more evolutionarily distant Holometabolous insect, the honeybee, *Apis mellifera* (*Amel*)[13], and in an Hemipteran (a recent outgroup to Holometabola), the milkweed bug *Oncopeltus fasciatus* (*Ofas*)[14]. In both species, *prd* is expressed before *gsb*, as is true in *Drosophila* and *Tribolium*[6,15]. Together, these results support the notion that Prd and Gsb took on their distinct segmentation roles, with Prd serving as the pair-rule gene, long before the evolution of *Drosophila*.

In light of the evolutionary conservation of *prd* as a pair-rule gene, it was surprising that *prd* was predicted to be absent in early versions of mosquito genomes and transcriptomes[16]. This loss was particularly puzzling because of Prd's essential role in segmentation and embryonic viability in both closely related Dipterans (flies) as well as other more distantly diverged insects, as mentioned above. We hypothesized that the subfunctionalization of *prd* and *gsb*, established prior to the divergence of Diptera, could have been reversed in mosquito lineages, resulting in redundancy and allowing for *prd* gene loss. To test this we analyzed the expression and function of *gsb* in the Indian malaria vector mosquito, *Anopheles stephensi* (*Aste*). We find that an expansion of *gsb* expression into *prd*'s typical spatiotemporal domain in *Anopheles* provides a possible mechanism for loss of *prd* from mosquito genomes. We then demonstrate that *gsb* has taken on the functionality of *prd* through analysis of a *gsb* loss-of-function mutant line, that we generated by CRISPR-Cas9. Our *Aste-gsb* mutant embryos display typical pair-rule defects with loss of alternate morphological segments and expression of alternate *en* stripes, characteristic of *Dmel-prd* and *Tcas-prd* mutants. Thus, in mosquitoes, *gsb* has seamlessly replaced *prd* in the segmentation gene network.

## Results

**Mosquitoes do not have a *prd* ortholog.** To test whether *prd* was lost before the divergence of the Dipteran clade Culicidae (mosquitoes), we used *Dmel*-Prd's conserved sequence regions[17] as the query in tblastn searches of *Aedes aegypti* (*Aaeg*), *Anopheles gambiae* (*Agam*), *Aste*, and *Culex quinquefasciatus* (*Cqui*) genomes. Significant hits were aligned to known insect (*Dmel*, *Tcas*, and *Amel*) and mouse (*Mus musculus* (*Mmus*)) Pax3/7 and Pax6 sequences and subjected to phylogenetic analysis. Phylogenetic trees identified mosquito Pax3/7 genes as *gsb* and *gooseberry-neuro* (*gsb-n*) rather than *prd* (Fig. 1a), with the next best hits clustering with *Pax6*. In parallel, we used aligned Pax3/7 amino acid sequences as query for HMMER searches of 22 mosquito genomes, again identifying a maximum of two Pax3/7 orthologs (*gsb* and *gsb-n*) in each mosquito genome, along with more distantly related *Pax6* genes (Supplementary Fig. 1). This more complete search uncovered a previously undetected *Culex gsb-n* ortholog, but no other new Pax3/7 genes. In sum, neither method of identifying Pax3/7 genes uncovered a mosquito *prd* ortholog.

Signature sequence motifs further support the absence of *prd* from mosquito genomes. Mosquito Gsb and Gsb-n share paired domain and homeodomain sequences with Prd proteins, but are distinguished from Prd by the presence of an octapeptide motif found in all Gsb and Gsb-n proteins (Fig. 1b). The composition of the octapeptide motif further distinguishes between Gsb and Gsb-n (Fig. 1b, bold). Synteny also supports the conclusion that the only Pax3/7 genes in mosquito genomes are *gsb* and *gsb-n*. *gsb* and *gsb-n* are neighboring genes in the *Aste* genome (Fig. 2), as they are in the *Drosophila*, *Apis*, and *Agam* genomes[13], while *prd* is located outside of this cluster. Microsynteny surrounding *gsb* and *gsb-n* even extends to additional co-linear genes within Diptera. In sum, phylogenetics, identifying sequence motifs, and synteny all support classification of mosquito Pax3/7 genes as *gsb* and *gsb-n* but not *prd*.

The lack of a *prd* sequence must be a loss from mosquitoes rather than a gain in lineages leading to *Drosophila* because *prd* orthologs are present in more distantly related insects such as *Tribolium* and *Apis* (Fig. 1c). We performed phylogenetic analysis with sequences from all major arthropod lineages to determine when *prd* and *gsb* diverged to allow for inference of the ancestral Pax3/7 state and to trace the history of these genes' subfunctionalization process. We found that *prd* and *gsb* originated from a Pax3/7 duplication event that occurred by or before the emergence of Pancrustacea 530 million years ago[18], based upon

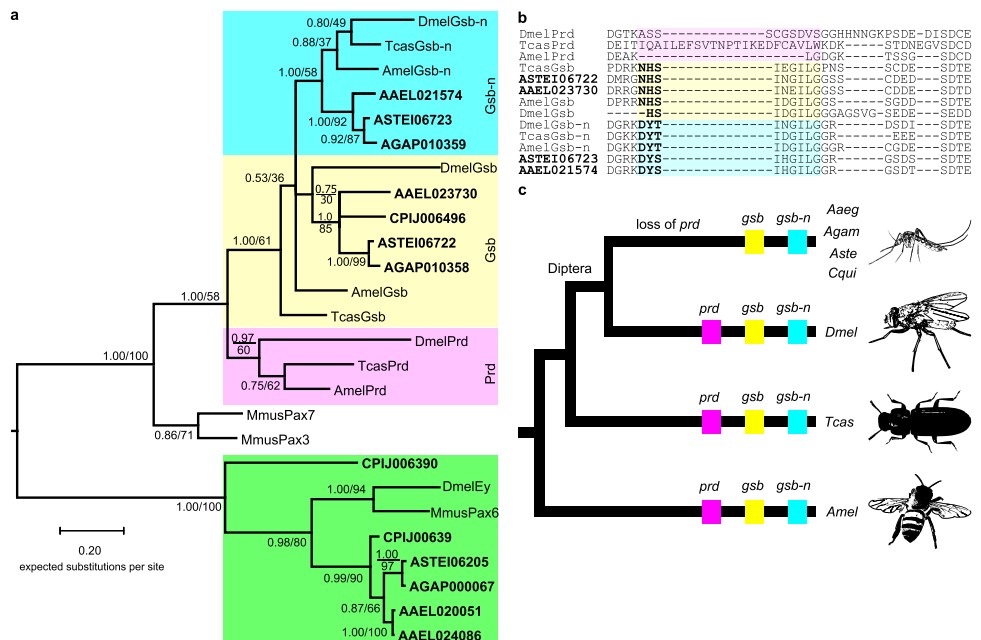

**Fig. 1 Mosquito genomes contain orthologs of *gsb* and *gsb-n*, but not *prd*. a** Phylogenetic tree built from previously characterized *Pax* genes and mosquito genome BLAST hits (bold) identifies orthologs of Gsb (yellow), Gsb-n (blue), and Ey (green). However, none of the mosquito hits cluster with known insect Prd sequences (pink). Values indicate statistical support for a node when tree topology is determined by MrBayes (posterior probability, above line) or PhyML (bootstrap support, below line). **b** Mosquito Pax3/7 sequences include Gsb-family signature motifs. Mosquito BLAST hits (bold) include either a Gsb-type (yellow) or Gsb-n-type (blue) octapeptide. Prd proteins do not contain an octapeptide and are not alignable in this region (pink). **c** *Pax3/7* repertoires in representative holometabolous insects. It can be inferred that *prd* was lost in the crown group of mosquitoes based on its presence in all other insects sampled. Species: *Aedes aegypti* (*Aaeg*/AAEL), *Anopheles gambiae* (*Agam*/AGAP), *Anopheles stephensi* (*Aste*/ASTEI), *Apis mellifera* (*Amel*), *Culex quinquefasciatus* (*Cqui*/CPIJ), *Drosophila melanogaster* (*Dmel*), *Mus musculus* (*Mmus*), *Tribolium castaneum* (*Tcas*).

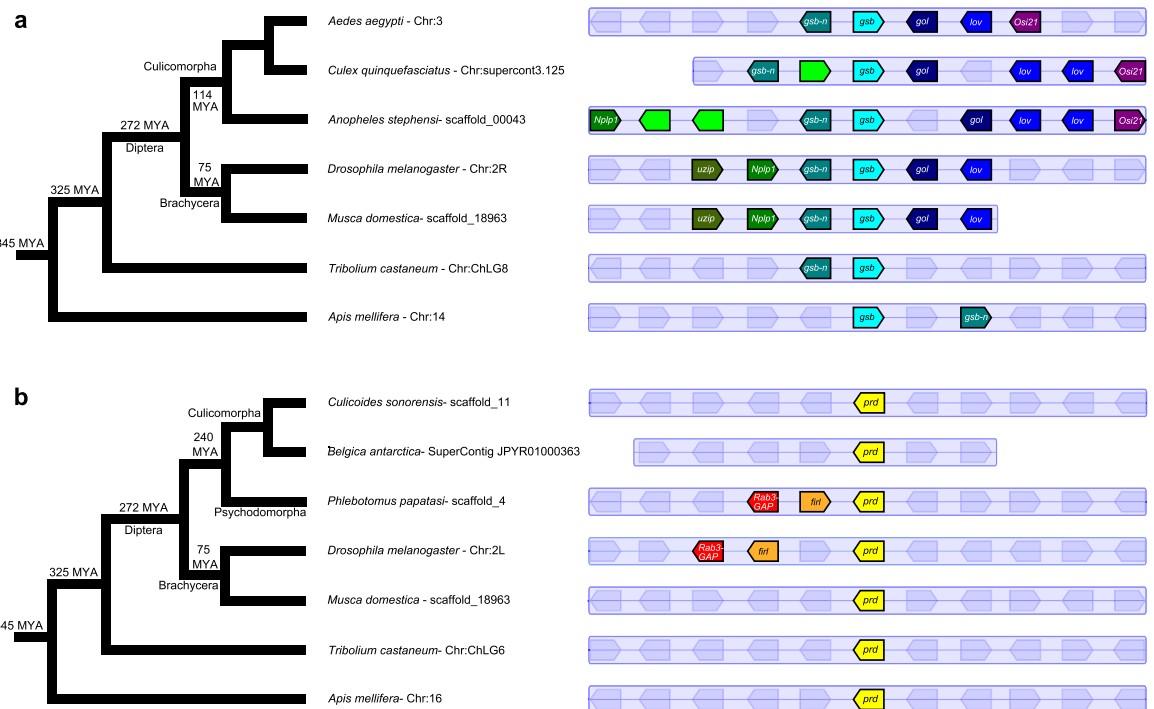

**Fig. 2 *gsb* and *gsb-n* are syntenic in insect genomes but *prd*'s closest neighbors are not conserved.** Microsynteny of insect *Pax3/7* genes was assessed using Genomicus Metazoa[52] and supplemented with additional species or newer assemblies with Ensembl Metazoa[53]. Orthologous genes are indicated by highlighting in the same color. Genes proximal to a *Pax3/7* gene without orthologs on the pictured scaffolds are shown in pale blue with no gene name label. Short scaffolds are shown as cut-off after the last predicted gene. **a** *gsb* (cyan) and *gsb-n* (dark cyan) are neighbors, or very nearly neighbors in diverse holometabolous insects. Within Diptera, neighboring genes on either side (*Nplp1* (green), *gol* (dark blue), and *lov* (blue)) also exhibit microsynteny, The conservation of this block of genes supports the orthology of mosquito *Pax3/7* genes and *Drosophila gsb* and *gsb-n*. **b** Orthologs of *prd* (yellow) exist in a wide variety of holometabolous insects, even in sister taxa of mosquitoes within Culicomorpha. However, there is little conservation of neighboring genes, even over short time scales.

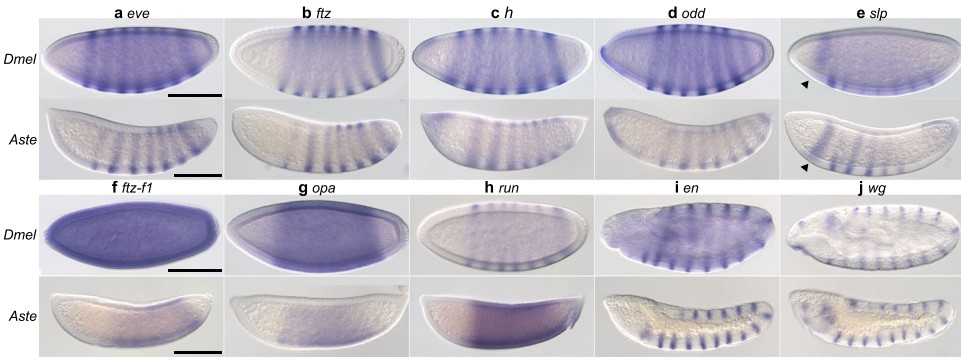

**Fig. 3 Expression of pair-rule and segment polarity genes suggests a highly conserved Dipteran segmentation gene network.** Gene expression assessed by whole-mount in situ hybridization. All embryos are shown as a lateral view with the anterior end on the left. **a–e** Pair-rule genes *eve, ftz, h, odd* and *slp* are expressed in a seven-stripe pattern at blastoderm in *Anopheles* (bottom row) that matches *Drosophila* expression (top row, shown for reference). For *h, odd,* and *slp*, other expression, such as the broad anterior *slp* stripe that precedes the pair-rule stripes (arrow heads), is also conserved. **f–h** Several *Anopheles* pair-rule gene orthologs (bottom row) exhibit broad expression at blastoderm. *Anopheles ftz-f1* expression is broad although not ubiquitous as in *Drosophila*. *opa* is expressed broadly throughout the trunk in both species. *Aste-runt* expression is expanded in interstripe regions, relative to *Drosophila* (top row). **i, j** Segment polarity genes *engrailed* and *wingless* exhibit highly conserved expression. Both are expressed in fourteen stripe patterns in both species at the extended germband stage. In all panels, scale bar = 100 μm. All panels in the same row were photographed at the same magnification and are shown at the same scale.

presence of both genes in crustacean, hexapod, and insect lineages (Supplementary Fig. 2). Further availability of myriapod sequences may later reveal an even earlier date of duplication. Examination of intron/exon structure of *gsb* and *prd* from representative Pancrustacea lineages reveals conservation in the general gene structures, including separation of sequences encoding the paired and homeodomains in different exons as well as conserved breakpoints within the paired domain sequence implicating tandem duplication in the initial gene duplication event (Supplementary Fig. 3). We find that no orthologs are composed of single exons, suggesting the genes did not diverge via retrotransposon[19,20] (Supplementary Fig. 3). Additionally, we examined the Dipteran lineage in more detail and found *prd* orthologs in Culicomorphan and Psychodomorphan flies that are more closely related to mosquitoes than to *Drosophila* (Supplementary Fig. 4). This indicates a recent loss of *prd* that is specific to mosquitoes within Culicomorpha. Retention of *prd* in many diverse dipteran genomes suggests that its essential gene functions were not redundant with *gsb*'s in the ancestor of this lineage. Thus, *prd* and gsb resulted from an ancient gene duplication event and *prd* was then lost at the base of the mosquito lineage, in spite of its conservation in the majority of insect lineages, including its sister clade within Diptera.

**Anopheles' embryonic segmentation gene network is largely conserved.** Several years ago, the Noll lab showed that expression of *gsb* under the control of *prd*'s *cis*-regulatory elements in transgenic *Drosophila* rescued *prd* segmentation defects. Thus, even though it is not identical at the sequence-level due to several hundred million years divergence, Gsb showed the potential to carry out Prd-like functions[21]. Did this regulatory handoff happen in nature? Since Gsb shares DNA binding specificity with Prd, it could seamlessly integrate into a preexisting Prd-dependent segmentation gene network. No other components of the network would be required to reshuffle. Alternatively, other players in the segmentation network may have altered to take *prd*'s ancestral role. To determine whether reshuffling of regulatory gene expression explains the loss of *prd*, we analyzed the expression of other pair-rule genes and their regulatory targets, the segment polarity genes, in *Aste* embryos (Fig. 3). We observed little change overall when compared to the well-studied *Drosophila* segmentation gene network[22]. *Aste-even skipped (eve)* is

expressed in seven stripes along the trunk of the blastoderm stage embryo (Fig. 3a), in agreement with previous work in *Agam*[23]. *Aste-fushi tarazu (ftz), -hairy (h), -odd skipped (odd),* and *-sloppy paired (slp)* are also expressed in seven-stripe patterns at blastoderm in both species (Fig. 3b–e). *ftz transcription factor 1 (ftz-f1)* and *Aste-odd paired (opa)* are each expressed in broad domains in both species (Fig. 3f, g). *Aste-ftz-f1* expression is more restricted than the ubiquitous expression in *Drosophila*. However, this difference would have no functional consequence since Ftz-F1 is only active where co-expressed with its obligatory partner, Ftz, which is expressed in seven stripes in both species. *Aste-runt (run)* is an exception to this pattern of conservation; it is expressed in seven stripes in *Drosophila* but in a broad pattern more reminiscent of *opa* in *Anopheles* (Fig. 3h). This variation is of interest but would not explain the loss of *prd*. Collectively, these patterns suggest that essentially the same gene network is used to specify segments in *Anopheles* and *Drosophila* and that the regulatory environment experienced by downstream segment polarity target genes is extremely similar. In keeping with this idea, expression of segment polarity genes *en* and *wingless (wg)* in *Anopheles* (Fig. 3i, j) is segmental, as in *Drosophila* and other insects[24]. Rather than employing an entirely distinct segmentation mechanism, *Anopheles* has likely replaced *prd* with some comparable activator, leaving the rest of the network intact.

**Expression of *Aste-gsb* merges aspects of *Dmel-prd* and *Dmel-gsb*.** We hypothesized that *gsb*, present in all mosquito genomes analyzed, is what replaced the function of *prd*. This option is particularly parsimonious as Gsb and Prd share DNA binding domains such that Gsb has the potential to bind preexisting Prd binding sites in *cis*-regulatory elements, maintaining proper expression of all of its downstream genes, without multiple, coordinated changes in target *cis*-regulatory elements. However, this would require spatial and temporal shifts in *gsb* expression, to include earlier, pair-rule-like expression. Therefore, we compared the developmental expression patterns of *Aste-gsb* to those of *Dmel-prd*[4] and *Dmel-gsb*[5] (Fig. 4a). In keeping with our hypothesis, *gsb* expression begins earlier in *Anopheles* than in *Drosophila* and has taken on pair-rule-like features. Both *Aste-gsb* and *Dmel-prd* are first expressed in a single, anterior stripe. Then both develop an alternating pattern of pair-rule-stripes that are initially four-cells wide, but resolve into thinner two-cell wide

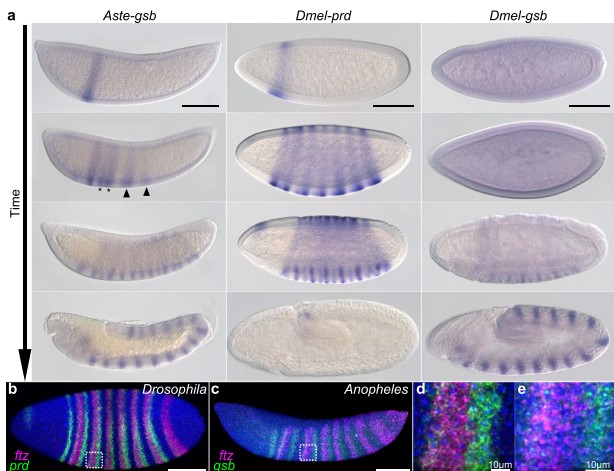

**Fig. 4 Early expression of *Aste-gsb* is reminiscent of *Dmel-prd*'s pair-rule pattern. a** Gene expression assessed by whole-mount *in situ* hybridization. Columns are labeled according to the dipteran *Pax3/7* gene analyzed. Embryo stages are in order of increasing time with the youngest at the top and the oldest at the bottom (arrow). *Aste-gsb* and *Dmel-prd* expression patterns are highly similar in the first three developmental stages shown. In the fourth row, *Aste-gsb* expression instead matches that of *Dmel-gsb*. **b**, **c** Double-fluorescent *in situs* reveal an alternating pattern of *ftz* expression (magenta) with *Dmel-prd* (**b**, **d** green) or *Aste-gsb* (**c**, **e** green). **d**, **e** are insets expanded from boxed regions shown in **b** and **c**, respectively. In both species, the indicated *Pax3/7* gene (green) is expressed in a faint, emerging stripe at the anterior of the *ftz* stripe, and again in a strong stripe abutting the posterior of the *ftz* stripe. All embryos are shown as a lateral view with the anterior end on the left. Scale bar = 100 μm, unless otherwise indicated.

stripes over time. Note that *Dmel-prd*'s seven pair-rule-stripes arise simultaneously, while *Aste-gsb* stripes develop in anterior to posterior order, as described in other holometabolous insects[13,25]. Newly emerging four-cell pair-rule-stripes can be seen more posteriorly (Fig. 4a, arrow heads and Supplementary Fig. 5) while older pair-rule-stripes have already started to split (Fig. 4a, asterisks and Supplementary Fig. 5). Next, *Dmel-prd* develops another seven stripes intercalated between the existing seven, creating a 14-stripe pattern as gastrulation begins. *Aste-gsb* is also expressed in 14 stripes at this stage. At this point, *Dmel*-Prd activates *Dmel-gsb* which carries on the 14-stripe segment polarity-type pattern through germband extension, where it functions to specify a portion of each segment[12]. *Dmel-prd* stripes are turned off as *Dmel-gsb*'s expression increases. However, *Aste-gsb* continues expression in 14 stripes through germband extension as well, suggesting conservation of segment polarity function. Thus, *Aste-gsb* appears to have taken on the pair-rule-expression pattern of *prd*, while retaining *gsb*'s segment polarity pattern.

In *Drosophila*, Prd is necessary for expression of odd-numbered *en* stripes, while Ftz activates even-numbered *en* stripes[11]. If Gsb were to replace Prd in the activation of downstream target genes, its position relative to Ftz would have to be maintained: this would ensure availability in the correct cells at the correct time to replace Prd's activation of downstream target genes. Further, as changes in expression of a transcription factor during evolution can have drastic, deleterious gain-of-function effects caused by ectopic expression of target genes, the precise expression of Gsb in only the cells that once expressed Prd is critical for successful transfer of function. We found that relative expression domains of *Aste-gsb* with *Aste-ftz* are nearly identical to those of *Dmel-prd* with *Dmel-ftz* (Fig. 4b–e). In both species, *ftz* stripes (magenta) alternate with *prd* or *gsb* (green) stripes. The posterior portion of each *ftz* stripe abuts a strong *prd*

or *gsb* stripe (Fig. 4d, e) with a single-cell wide region between the *prd* or *gsb* stripe and the next *ftz* stripe. Taken together, these analyses show that *Aste-gsb*'s newly acquired pair-rule-pattern is positioned appropriately in *Anopheles* embryos to take on Prd's role in activating downstream target genes.

**Aste-gsb loss-of-function mutants exhibit pair-rule phenotypes.** Expression at the appropriate place and time is a prerequisite for gene function; however, functional integration into an existing gene network could occur sometime after acquisition of a new expression pattern or never at all. To determine whether *Aste-gsb* performs the pair-rule gene network function fulfilled by *prd* in other insects, we used CRISPR-Cas9 to ablate *gsb* function, inserting *3XP3eGFP*[26] into its coding region, allowing for identification of insertion events by eye fluorescence (Fig. 5a). Insertion of *3xP3eGFP* into the *gsb* locus was validated by PCR followed by sequencing (Supplementary Fig. 6). Individuals with fluorescent eyes were crossed to wild-type mosquitoes to develop a line to study the effects of *gsb* loss on segment development and En expression. To control for potential CRISPR off-target effects, self-crossed heterozygous *gsb*[3xP3GFP] mosquitoes (*gsb* cross) were compared to a "control cross" composed of *gsb*[3xP3GFP/+] females mated to *gsb*[+/+] male siblings (Fig. 5b).

To assess effects of the *gsb* mutation on embryonic development, eggs from these crosses that failed to hatch were dissected to remove larvae for cuticle preparations. Larvae from control cross eggs exhibited a wild-type body plan composed of three thoracic (T1–3) and ten abdominal (A1–10) segments[27] (Fig. 5c). In contrast, in the majority of larvae dissected from unhatched *gsb* cross eggs, analysis of cuticles revealed pair-rule defects, in which alternate segments are absent. Segment identify can be determined in *Anopheles* by the patterns of hair-like setae and other cuticular structures. T2, which can be distinguished by two bunches of setae in wild-type cuticles, is always absent in *gsb*[−/−] pair-rule mutants (Fig. 5c, d). The remaining abdominal segments were identified as A2, 4, 6, 8, and 10 by defining features such as the presence of long setae normally on A1–3, which are only seen on the first remaining abdominal segment in pair-rule mutants, indicating that it is A2. A8 typically has two lateral comb-like cuticular structures and A10 has very long setae at the tip. Both of these segments are still observed in *gsb*[−/−] pair-rule mutants (Fig. 5e, f). However, A9, which is reduced to a spiracle or respiratory siphon in mosquito larvae[28] is notably absent in pair-rule mutants. Importantly, larvae that fail to hatch and exhibit pair-rule phenotypes represent 25 ± 3% of the eggs collected from *gsb* crosses, as expected if they are the *gsb*[3xP3GFP/3xP3GFP] individuals, which is significantly different from the control crosses where no pair-rule phenotypes were observed ($p = 0.0015$, Student's *t* test). The genotype of *gsb*[3xP3GFP/3xP3GFP] individuals was confirmed by PCR of genomic DNA from dissected pair-rule mutant larvae (Supplementary Fig. 6). This is the first report of a pair-rule mutant in mosquitoes to our knowledge, and it resulted from mutating a gene that has no pair-rule function in other insects studied to date.

As mentioned above, a major role of fly and beetle Prd is to regulate the expression of alternate *en* stripes. *Aste-gsb*[3xP3GFP] homozygotes exhibit defects in En expression characteristic of pair-rule mutants in other species[9,11], in which only seven out of fourteen stripes are present (Fig. 5g, h). Consistent with straightforward Mendelian genetic expectations, this seven-stripe phenotype was observed in 25 ± 3% of *gsb* cross progeny but 0% of control cross progeny ($p = 0.0114$, Student's *t* test). In sum, *Aste-gsb* loss-of-function mutants exhibit pair-rule cuticle phenotypes and loss of alternate En stripes. Importantly, both of these phenotypes are characteristic of *Dmel-prd* mutants, but not

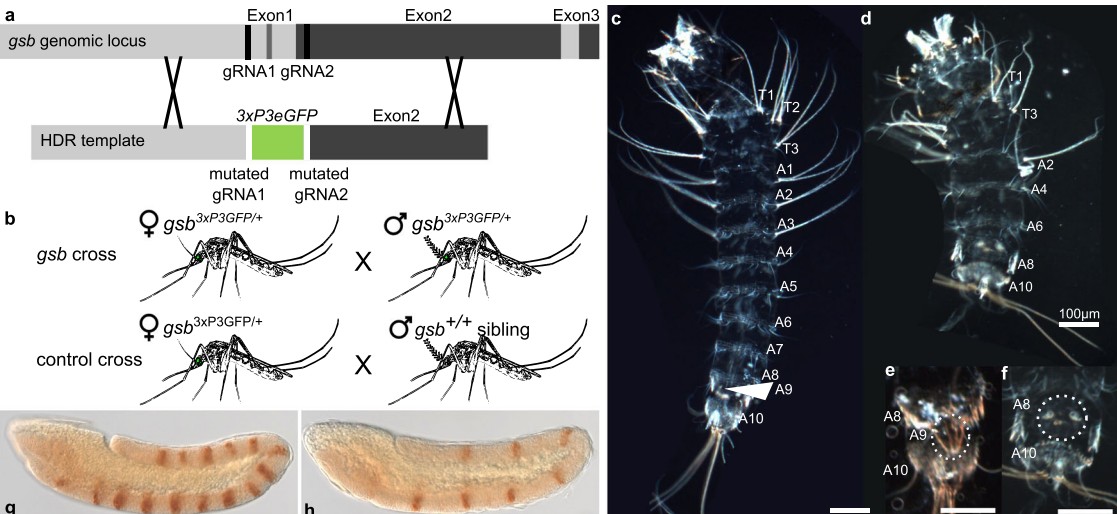

**Fig. 5 Mutation of *Aste-gsb* results in pair-rule phenotypes. a** Schematic illustrating gene-editing approach to introduce *3xP3GFP* into the coding region of *Aste-gsb* by homology directed repair. **b** Strategy for generating *gsb* homozygous mutant embryos and controlling for off-target effects generated by CRISPR-Cas9. Heterozygous *gsb^3xP3GFP/+* female mosquitoes are selected by *GFP* expression and mated to either their *gsb^3xP3GFP/+* male siblings (*gsb* cross) or *gsb^+/+* male siblings (control cross). **c–f** Hatch rates and representative larval phenotypes were determined by examining all progeny from four biologically distinct replicate cross experiments. **c** Representative first instar larvae dissected from a control cross egg that failed to hatch. Body pattern of three thoracic (T1–T3) and ten abdominal (A1–A10) segments is typical of wild-type mosquito larvae. **d** Representative larvae dissected from a *gsb* cross. Alternate segments are deleted with respect to the wild-type larvae (**c**). **e** Posterior-most segments of a control cross larvae. A9 is a respiratory spiracle rather than a typical body segment (circled). **f** The posterior-most segments of a *gsb* mutant reveal retention of A8 and A10, but loss of A9. **g, h** Representative En expression phenotypes were determined by examining all progeny from three biologically distinct replicate cross experiments. **g** A control cross germband-stage embryo immunostained for En protein expression (brown). Fourteen stripes are evident at this stage. **h** a *gsb* mutant embryo with only seven En stripes. In **g, h** embryos are shown in a lateral view with the anterior oriented to the left. In all panels, scale bar = 100 µm.

of *Dmel-gsb* mutants[12,29]. This suggests that loss of *prd* from mosquito genomes was possible because *gsb* performs its essential functions.

## Discussion

Together, these results demonstrate one evolutionary mechanism underlying the loss of an essential gene during evolution that obviates detrimental consequences to the organism or species. After gene duplication creates paralogs (such as *prd* and *gsb*), additional changes to the genome that promote the preservation of both gene copies often occur. In cases of subfunctionalization, paralogs divide gene expression patterns amongst themselves, avoiding redundancy. *Dmel-prd* and *Dmel-gsb* provided the first experimental validation that paralogs can evolve divergent functions through alterations in their *cis*-regulation rather than protein function[3]. That work predicted that paralogs with shared protein function could become functionally equivalent if they later evolved overlapping gene expression patterns. This functional equivalence would allow for loss of one paralog, as they would have become redundant. Gitelman proposed the term "synfunctionalization" for this gene loss mechanism in which subfunctionalization is essentially reversed (Fig. 6a)[30]. It is fitting that here, we have provided the first functional validation of synfunctionalization to our knowledge by again comparing *prd* and *gsb*.

The initial subfunctionalization of *prd* and *gsb* is likely ancient, as opposed to a phenomena unique to *Drosophila* development. As is true for *Drosophila*, *prd* is expressed prior to *gsb* in the flour beetle, *Tcas*[6,15], the honeybee, *Amel*[13], and the milkweed bug *Ofas*[14] (schematized in Fig. 6). Other features differing between *Dmel-prd* and *Dmel-gsb* are also conserved in these species, such as initially wide stripes of *prd* in *Tribolium* and *Apis* compared to the lingering *gsb* stripes in every segment at later stages. These

observations suggest that subfunctionalization to pair-rule vs segment polarity roles for *prd* vs *gsb* observed in *Drosophila* was established at least by the emergence of the lineage Holometabola.

The most evolutionarily distant lineage from Holometabola where both *prd* and *gsb* have been assessed is the hemipteran, *Ofas*. In this organism, none of the classic *Drosophila* pair-rule genes, including *prd*, have pair-rule function, even though pair-rule patterning is a part of its development[14,31]. Even so, *Of-prd* is expressed in each newly emerging segment, while *Of-gsb* is expressed later, when the segments are established (see Fig. 6), suggesting an origin of *prd* and *gsb* subfunctionalization that could even predate *prd*'s role in pair-rule patterning. The initial temporal subfunctionalization of *prd* and *gsb* may have occurred even earlier than that, soon after a gene duplication event in Pancrustacea (Fig. 6a, Supplementary Fig. 2b). In long-diverged non-insect arthropods and arthropod outgroups, without clear orthologs to both *prd* and *gsb*, *Pax3/7* genes are expressed during segmentation, usually in a segment polarity-type pattern, but pair-rule-like expression, has also been observed[32–36]. The only characterized crustacean *gsb* ortholog is expressed in segment polarity stripes that matches expression in insects lending support to the idea that *prd* and *gsb* subfunctionalized soon after their duplication (Supplementary Fig. 2)[37]. Together these data strongly support that idea the ancestral *Pax3/7* that generated *prd* and *gsb* had a role in segmentation, and suggest that dual roles in pair-rule as well as segment polarity patterning that are plausible. Rapid partitioning of these functions among paralogs upon duplication is also plausible. However, it is still unclear whether pair-rule patterning emerged once, or as multiple, convergent events[14], and therefore the initial subfunctionalization may have been a simple partitioning of early vs late expression (as in *Oncopeltus*). In light of the 530 million years divergence of *prd* and *gsb*, their retained ability to undergo synfunctionalization is amazing[18].

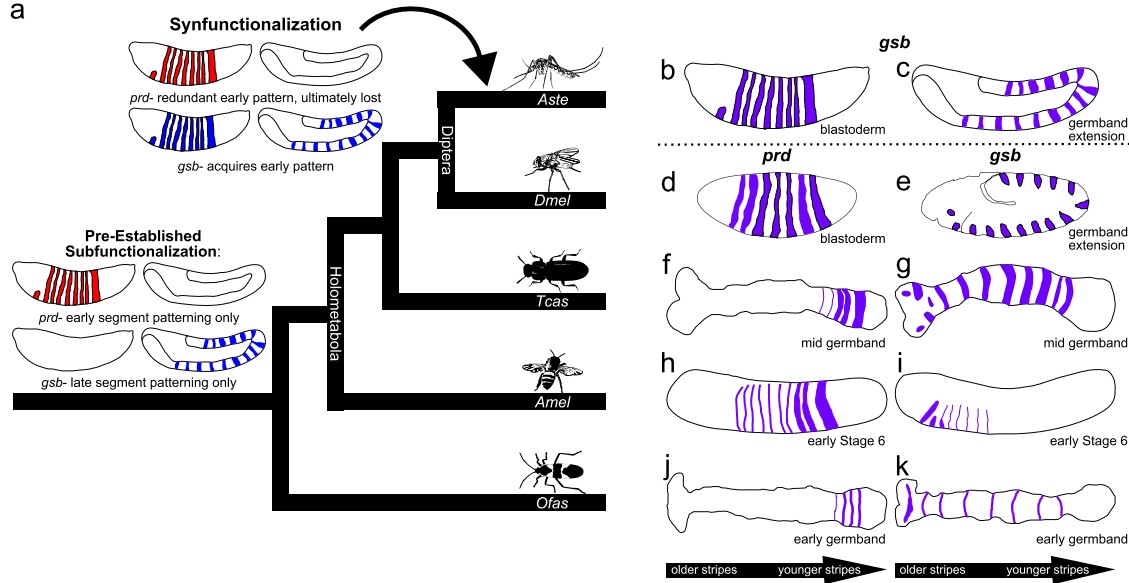

**Fig. 6 Comparison of *prd* and *gsb* expression patterns in insects reveals ancient subfunctionalization and recent synfunctionalization.** Available literature on insect *prd* and *gsb* expression was reviewed and synthesized with our results into schematic format. **a** Cladogram depicting the relationships between species used in this comparison, with schematics indicating the synfunctionalization mechanism and evolutionary timing. Four members of Holometabola (*Aste*—*Anopheles stephensi*, *Dmel*—*Drosophila melanogaster*, *Tcas*—*Tribolium casteneum*, and *Amel*—*Apis mellifera*) representing divergent orders (Diptera, Coleoptera, and Hymenoptera, respectively) were compared as well an outgroup, Hemiptera (*Ofas*—*Oncopeltus fasciatus*). Gene expression patterns from each species are shown immediately to the right of the matching cladogram position. Arrows underneath in situ drawings denote segments added in anterior to posterior order in sequentially segmenting species (**f**–**k**), such that in similarly staged embryos, stripes closer to the head (shown on the left) are older than stripes closer to the posterior end (shown on the right). **b**–**e** Representative drawings of results shown in Fig. 4 and Supplementary Fig 5. **b**, **c** *Aste-gsb* expression; **d**, **e** *Dmel-prd* and *Dmel-gsb* peak expression patterns, respectively. **f** *Tcas-prd* expression at mid-germband stage in stripes emanating from the segment addition zone, and fading upon segment establishment, based on Choe and Brown[6]. **g** *Tcas-gsb* expression in established segmental units along the trunk of the mid-germband-stage embryo, based on Aranda et al.[15]. **h**, **i** Expression of *Amel* genes in early stage 6 embryos based upon Osborne and Dearden[13]. **h** Expression of *Amel-prd* as broad pair-rule like stripes upon emergence that resolve into thinner stripes with age. **i** Expression of *Amel-gsb* as thin stripes that appear in older segments only. **j**, **k** Expression of *Ofas* genes in early germband-stage embryos based upon Reding et al.[14]. **j** Expression of *Ofas-prd* in thin stripes emanating from the segment addition zone that fade as segments mature. **k** Expression of *Ofas-gsb* as a thin stripe in each established segmental unit.

Our analysis suggests that the original gene duplication that created *prd* and *gsb* did not occur by retrotransposon (Supplementary Fig. 3). This leaves open the possibility that *cis*-regulatory elements were also duplicated and then modified slightly to create the temporal sequence of *prd* expression followed by *gsb* expression observed in many insect lineages (Fig. 6). In *Drosophila*, this temporal order of expression is ensured by Prd's direct activation of *gsb*[38]. Shifting expression dependence to a new activator could simultaneously allow earlier *gsb* expression and reduce the need for Prd, as *gsb* is one of Prd's few known target genes. Additional work on *cis*-regulatory elements in mosquitoes as well as functional work in a wider variety of Dipterans would enable us to test these ideas.

Here, we have shown that *Aste-gsb* is not only expressed like *Dmel-prd*, it also phenocopies *Dmel-prd*'s mutant phenotype. Our work demonstrates that the loss of *prd* from mosquito genomes had a neutral impact on the highly conserved process of segmentation. The ability to substitute *prd*'s paralog *gsb* into the segmentation gene network confers important stability in body plan patterning. There is evidence that this type of substitution may have occurred repeatedly within insects, with an inversion in which *gsb* was lost but *prd* acquired *gsb*-type expression occurring in the jewel wasp, *Nasonia vitripennis*[39]. It would be interesting to learn whether re-evolved redundancy with *gsb* also allowed *prd* to be lost in Lepidopterans (Supplementary Fig. 2).

More generally, synfunctionalization could be a pervasive gene loss mechanism based on the wide variety of organism and gene families that show tantalizing gene expression-level evidence of this phenomenon[30,40,41]. Gene loss has long been considered a neutral process with little ability to cause adaptive evolutionary change, but this view is rapidly changing as genomic data reveals the extent of gene loss that has occurred over the course of evolution[42]. Insects in particular have undergone large amounts of gene loss and are also among the most diverse and species-rich clades of organisms on the planet. This may not be a coincidence.

## Methods

**Gene identification and phylogenetic tree construction.** To identify mosquito *Pax3/7* genes, the *Dmel*-Prd paired domain+homeodomain sequence (Genbank Accession M14548.1) (conserved amino acids 33–218) was used as a query for a tblastx search of the *Aaeg* Liverpool AaegL5, *Agam* PEST AgamP4, *Aste* Indian AsteI2, and *Cqui* Johannesburg CpipJ2 genome assemblies housed on Vectorbase[43]. Hits with an *E* value of 1e − 30 or less were aligned to known insect (the flour beetle, *Tcas* and honeybee *Amel*) and mouse *Mmus* Pax3/7 and Pax6 sequences (see Supplementary Table 1) using MUSCLE[44] and subjected to phylogenetic analysis to determine orthology. Phylogenetic trees were made in TOPALI v2.5 build 13.04.03 (www.topali.org) with MrBayes v3.1.1 algorithm (Parameters: Runs: 2, Generations: 100,000, Sample Freq.: 10, Burnin: 25%)[45,46]. A second, complimentary tree was produced in TOPALI v 2.5 using a maximum likelihood approach, PhyML-aLRT (Version 2.4.5), with 100 bootstrap runs[47]. Both trees were constructed using a JTT + gamma substitution model as recommended by model selection analysis in TOPALI (Fig. 1a). Schematic topology is based on Misof et al.[18].

Additionally, mosquito genomes were searched using an alternate method. Known Pax3/7 amino acid sequences were aligned using MUSCLE:[44] *Mmus* (mouse) Pax3 and Pax7, *Dmel*-Prd, *Tcas*-Prd, *Amel*-Prd (see Supplementary Table 1 for accession numbers). This alignment was used as a search query for hmmsearch (EMBL-EBI HmmerWeb version 2.33.0, https://www.ebi.ac.uk/Tools/hmmer/search/hmmsearch)[48], using the "reference proteomes" database and restricted to taxa ID 7157 (mosquitoes). *E* value cutoffs were left at defaults

(seq = 0.01, hit = 0.03). This search included all available mosquito genomes (19 *Anopheles* species, 2 *Aedes* species, and *Cqui*). 560 hits were obtained. A complete list of all hits in FASTA format was downloaded and run through NCBI conserved domains batch web search tool (https://www.ncbi.nlm.nih.gov/Structure/bwrpsb/bwrpsb.cgi)[49]. Any sequence that had even a partial match to a Pax domain plus a match to even a partial homeodomain was added to a new FASTA document.

The hmmsearch and conserved domains filtering was repeated using taxa ID 7227 (*Dmel*) so that all isolated sequences could be compared to known *Dmel* genes after phylogenetic tree building. Filtered *Dmel* hits were added to the mosquito filtered dataset. The following previously studied Pax3/7 sequences were also added to the filtered dataset to serve as outgroups: *Mmus*-Pax3, *Mmus*-Pax7, *Tcas*-Prd, *Tcas*-Gsb, *Tcas*-Gsb-n, *Amel*-Prd, *Amel*-Gsb, *Amel*-Gsb-n (see Supplementary Table 1 for accession numbers). Then, the complete list of FASTA sequences was aligned using MUSCLE[44]. The sequences were trimmed to match the aligned regions and used to build phylogenetic trees. Trees were constructed as described above using MrBayes v3.1.1 algorithm (Parameters, Runs: 2, Generations: 250,000, Sample Freq.: 50, Burnin: 40%) and PhyML-aLRT (100 bootstrap runs) both with JTT + gamma substitution models (Supplementary Fig 1).

A tree to determine the divergence time of *prd* and *gsb* was constructed using the same methodology as the original BLAST Pax3/7 search described above, except that genomes from divergent arthropod lineages (Chelicerata, Myriapoda, Crustacea), representatives within Hexapoda (e.g. Collembola, Polyneoptera, Hemiptera, and Holometabola) and an arthropod outgroup (Onychophora) were searched. We attempted to keep the coverage per lineage relatively even as there are far more genomes available for some lineages (e.g. Holometabola) than for others (e.g. Myriapoda). See Supplementary Table 2 for sequence accession numbers and sources of genomic data. MrBayes (Runs: 2 Generations: 300,000, Sample Freq.: 35, Burnin: 40%) and PhyMl (100 bootstrap runs) and RaxML (Version 2.2.3)[50] (100 bootsrap runs) trees were constructed using a JTT + gamma substitution model (Supplementary Fig 2). Nodes were considered well-supported if validated by both MrBayes and one of the two ML methods. Schematic tree topology is based on Misof et al.[18]. Sequences obtained from Ensembl Metazoa (release 47—April 2020) representing a variety of Pancrustacea lineages were viewed under "region in detail" for analysis of exon/intron boundaries and locations of BLAST hits to *Dmel-prd* (M14548.1). This information was used to draw the schematics in Supplementary Fig 3.

Similarly, a tree of Dipteran sequences to determine the timing of *prd* loss was constructed using the same methodology as the original BLAST Pax3/7 search described above, except that genomes for representative Nematoceran (i.e., non-Brachyceran) fly lineages were searched. See Supplementary Table 1 for sequences used and sources of genomic data. MrBayes (Runs: 2 Generations: 500,000 Sample Freq.: 15 Burnin: 40%), and PhyMl (100 bootstrap runs) trees were constructed using JTT + gamma substitution model (Supplementary Fig 4). Schematic tree topology is based on Wiegmann et al.[51].

Microsynteny of insect *Pax3/7* genes was assessed using Genomicus Metazoa (web-code version: 2014-07-06, database version: 30.01, https://www.genomicus.biologie.ens.fr/genomicus-metazoa-30.01/cgi-bin/search.pl)[52]. This analysis (Fig. 2) was supplemented with additional species or newer assemblies by examining the orthology of neighboring genes in Ensembl Metazoa[53]. See Supplementary Table 3 for gene information.

*Aste* orthologs of *Drosophila* segmentation gene network genes were identified by reciprocal BLAST[54]. First, CDS sequences corresponding to the Gene IDs listed in Supplementary Table 4 for each *Drosophila* gene were obtained from FlyBase[55] and used as queries in a tblastx search against the *Aste* Indian strain genome v2[56] and AsteI2.3 gene set housed on Vectorbase[43]. All hits with an *E* value of 1e − 30 or less were BLASTed against the *Drosophila* genome v6.13 housed on FlyBase to identify the *Anopheles* ortholog of the desired segmentation gene.

**Mosquito rearing.** A starter colony of *Aste* was obtained from the IBBR Insect Transformation Facility (University of Maryland, Shady Grove). The mosquitoes were maintained at 29 °C with 80% humidity and a 12-h light/dark cycle. Larvae were fed a diet of powdered dog chow (Purina) or fish food (Tetramin). Adults were provided with 10% (wt/vol) sucrose solution ad libitum. Bovine blood treated with sodium heparin anticoagulant (Lampire, Pipersville, PA) was provided by artificial membrane feeder 3–5 days prior to oviposition. Transgenic mosquito lines were housed separately in a BSL-2 facility, but reared under the same conditions. Transgenic mosquitoes were housed, contained, and disposed of in accordance with protocols approved by the University of Maryland's Institutional Biosafety Committee.

**Generation of transgenic lines.** To enable the rapid identification of mutant individuals for genetic crosses we used Homology Directed Repair (HDR) to disrupt the *Aste-gsb* coding region and insert *3XP3GFP* into the region normally occupied by the first exon of *Aste-gsb*, generating the allele *gsb^{3xP3GFP}* (Fig. 5a). We implemented CRISPR-Cas9 with HDR using gene-editing protocols established for *Aaeg* and *Agam*[57,58]. Briefly, two gRNAs were designed to target the region surrounding the first exon using CHOPCHOP v2[59] (gRNA1, 5′ATTGTCACACA GCACAACCCAGG3′, gRNA2, 5′ACTGCGCATCGTCGAGATGGCGG3′). gRNAs were synthesized according to established protocols (https://www.crisprflydesign.org/grnatranscription/), using the T7 MEGAscript kit

(ThermoFisher Scientific) for in vitro transcription. Homology arms for the HDR construct included 900 bp of sequence from immediately upstream of gRNA1 and 800 bp of sequence from immediately downstream of gRNA2. These were fused to the 5′ and 3′ ends of a *3xP3GFP* cassette through PCR. The fused PCR product was inserted into pGEM-T easy (Promega, Madison, WI). Injection mix was made up of 40 ng/µL each gRNA, 300 ng/µL HDR donor plasmid, and 250 ng/µL Cas9 protein (PNA Bio, Newbury Park, CA) in standard injection buffer (5 mM KCl, 0.1 mM NaPO4, pH 6.8[60]). This mix was sent to the Insect Transformation Facility (Institute for Bioscience and Biotechnology Research, University of Maryland) for injection into *Aste* India strain and establishment of a transgenic line. Positive transformants were selected by visualizing neural GFP expression[26]. Insertion of *gfp* into the *gsb* coding region was confirmed by sequencing a PCR product that included the entire HDR region plus some surrounding genomic sequence to ensure that the repair vector had not integrated elsewhere in the genome. All cloning and confirmation primers are listed in Supplementary Table 5.

GFP+ individuals were outcrossed to wild type (India strain) to propagate the line, or self-crossed for experiments. Both outcross and self-crosses contained as many mosquitoes as were available, typically dozens of each sex, but a minimum of ten individuals of each sex, to promote mating and oviposition behaviors. Each set of experimental self-crosses represents a different generation and therefore, a different number of outcrosses. Two outcrosses were performed before experimental data were collected, but most replicates represent 5–6 outcrosses. To control for potential off-target effects introduced into the line during gene editing, experimental self-crosses were composed of *gsb^{3xP3GFP}/+* females mated to either their *gsb^{3xP3GFP}/+* (*gsb* cross) or *gsb^{+/+}* (control cross) male siblings. These *gfp*-negative male siblings are just as likely to contain mutations due to nonhomologous end joining at other loci as *gfp +* males and allowed us to determine whether phenotypes were due to knocking *gfp* into the *gsb* locus versus other undetectable mutations in the genetic background.

**Embryo fixation and analysis of gene expression.** *Drosophila w^{1118}* strain embryos were collected and fixed for in situ hybridization and antibody staining as previously described[61,62]. Briefly, embryos were dechorionated in household bleach, diluted 50:50 with tap water for 3 min, then rinsed thoroughly with water. Then, the embryos were fixed in a 1:1 solution of 4% PFA in PBST and heptane for 20 min with vigorous shaking. Embryos were freed from their vitelline membranes by shaking them in a 1:1 solution of heptane and methanol, then rinsed several times in methanol and stored at −20 °C until use.

*Anopheles* embryos were collected over the course of 2 h on wet filter paper, then removed from the cage and aged an additional 3–4 h to obtain desired stages. *Aste* embryos were fixed essentially as described for *Anopheles gambiae*[23,63]. To remove the exochorion, the embryos were incubated in 1.3% sodium hypochlorite for 75 s. The embryos were then placed in glass scintillation vials with a 1:1 mixture of heptane and 4% formaldehyde in PBST, and then rocked gently on nutator for 25 min. Afterwards, the formaldehyde phase was removed with a glass Pasteur pipette, replaced with deionized water and the tube was gently inverted to wash. The water phase was then replaced once more, and the embryos were rocked gently for an additional 30 min on the nutator. The water phase was then removed, and scintillation vials were filled to the top with boiling deionized water and incubated for 30 s. The hot water phase was quickly removed and replaced with fresh deionized water prechilled on ice. Vials were then placed on ice for an additional 15 min. All liquid was then completely removed, and the heptane phase was replenished with fresh reagent. To crack the endochorion, an equal volume of methanol was added, and the vials were strongly swirled once to break the clumps of embryos. Vials were allowed to stand for another 10–15 min, and then the heptane and methanol phases were removed and the embryos were washed several times with methanol. The embryos were stored in methanol at −20 °C until use. Just before use, embryos were stuck to toupee tape, submerged in cold 70% ethanol and peeled from their endochorions using needles from 1-ml insulin syringes.

For both species, in situ hybridization was performed essentially as described but with a few modifications[61,62]. Fixed embryos were rinsed in methanol twice and postfixed for another 25 mins in 4% paraformaldehyde/PBST. In place of Proteinase K treatment, embryos were heated at 95 °C for 5 mins. Next the embryos were pre-hybridized for an hour at 60 °C in hybridization buffer (50% formamide, 5× saline-sodium citrate, 100 µg/mL salmon sperm DNA, 100 µg/mL heparin, 0.1% Tween-20). Antisense RNA probes were synthesized using digoxigenin or biotin RNA labeling mix (Roche), diluted in hybridization buffer (typically 1:1500) and added to the pre-hybridized embryos for overnight incubation at 60 °C. The next day, the probe was removed by washing 1× in hybridization buffer and 1× in 50:50 hybridization buffer/PBST at 60 °C, followed by 4 washes in PBST at room temperature. Embryos were then incubated with anti-DIG-AP (1:2000, Roche) for 1–2 h at room temperature, followed by 4 × 30 min washes in PBST and 1× wash in staining buffer (0.1 M Tris pH 9.5, 0.1 M sodium chloride, 50 mM magnesium chloride, 0.1% Tween-20). Finally, patterns were detected by NBT/BCIP color reaction (staining buffer plus 450 µg NBT and 175 µg BCIP). Fluorescent in situ hybridization experiments only differed in that anti-DIG and anti-biotin were HRP conjugated (1:2000, Jackson ImmunoResearch, West Grove, PA) and TSA Plus Cyanine 3 and Fluorescein System (Perkin Elmer, Waltham, MA) was used according to the manufacturer's instructions. TSA reaction length was determined empirically for best signal to noise ratio. Reactions

were ultimately performed for 5 min for *Drosophila* samples, but 15 for *Anopheles* samples. Fluorescent in situs were co-stained with DAPI (1:10,000, Sigma). Primer sets and cDNA clones used in probe synthesis are listed in Supplementary Table 4. cDNA clones were obtained from the Berkeley *Drosophila* Genome Project *Drosophila* Gene Collection[64]. Antibody staining was performed according to established protocols[65]. Briefly, fixed embryos were washed 3× in PBST and incubated in 1:5 anti-Engrailed antibody overnight at 4 °C (catalog number 4D9, Developmental Studies Hybridoma Bank). This antibody was validated for use in a mosquito of the same genus in a previous study[66]. The following day, the embryos were washed 3× in PBST and then incubated with 1:200 biotinylated goat anti-mouse secondary antibody for 1–2 h at room temperature (BA9200, Vector Labs). After 3× washes in PBST plus an additional overnight wash to remove background, the embryos were incubated for 1 h in ABC reagent (Vectastain Elite ABC kit, Vector Labs), then washed 3× in PBST again. Finally, expression was detected by DAB reaction (SigmaFast reagent, Sigma-Aldrich). Colormetrically stained embryos were visualized using DIC on a Zeiss Axio Imager.M1 microscope while fluorescent in situs were imaged on a Zeiss LSM710 confocal microscope. Uniform adjustments were applied to images for clarity in ImageJ[67].

**Larval hatch rates and cuticle preparation.** *Anopheles* embryos were collected overnight on wet filter paper, then removed from the cage and aged for two days at 29 °C. Exochorions were then removed with 1.3% sodium hypochlorite for 75 s. Eggs were allowed to adhere to a strip of toupee tape stuck to a plastic petri dish lid and submerged in RO water. Eggs were scored as "hatched" if the eggshell appeared deflated and/or had a large opening near the anterior. For eggs that were not scored as hatched, the remainder of the eggshell was removed manually with a 25 gauge syringe needle. Eggs that did not contain larvae were not scored. Larvae that failed to hatch were scored based on the number of abdominal segments and clusters of setae on the thorax. Freed larvae were collected and incubated in 1:4 glycerol/acetic acid overnight at 60 °C, then transferred to a slide, covered in lactic acid, and incubated for an additional day before being analyzed under dark field microscopy (Leica DMRB). Uniform adjustments were applied to images for clarity in ImageJ[67].

**Statistics and reproducibility.** Experiments on transgenic embryo samples were repeated multiple times by setting up equivalent experimental and control crosses with subsequent generations of the same CRISPR-induced line. Samples were not random as they each contained embryos from a pool of sibling insects. Experimental groups were carefully compared to control groups, also composed of sibling insects from the same rearing pan as the experimental group. Both crosses in a replicate experiment were constructed from the same pool of insects to control for number of outcrosses to wild type and overall genetic background, in addition to exact rearing conditions, which affect fecundity. No sample size calculation was performed in advance. All embryos obtained from each timed collection were examined. No data were excluded.

Significance of the increase in seven-stripe pattern embryos in the *gsb* vs control cross determined by paired two-tailed student's *t* test. Antibody staining experiment was replicated three times with distinct biological samples. Each replicate experiment was made up of *gsb* and control cross samples that were collected, fixed and stained in parallel. Each sample contained 39–88 germband-stage embryos. For control samples, the mean 14-stripe pattern was 100%, with a standard deviation of 0. For *gsb* cross samples, the mean 14-stripe pattern was 75.2% with a standard deviation of 4.6. Test results: $t = 9.2920$, $df = 2$, $p = 0.0114$, 95% CI = 13.316–36.284. The effect size was calculated as Hedge's *g* (1.68).

Each hatch rate sample consisted of a minimum of 30 eggs, but more typically contained 100–300 eggs. Four biological replicate hatch rate experiments were performed. Statistical significance was determined by paired two-tailed student's *t* test.

For control samples, the mean hatch rate was 93.6% (0.936), with a standard deviation of 0.058. For *gsb* cross samples, the mean hatch rate was 67.1% (0.671) with a standard deviation of 0.057. Test results: $t = 11.2905$, $df = 3$, $p = 0.0015$, 95% CI = 0.190538–0.340112. The effect size was calculated as Hedge's *g* (61.2)

**Reporting summary.** Further information on research design is available in the Nature Research Reporting Summary linked to this article.

## Data availability

No new datasets were generated or analyzed by the current study. Accession numbers for gene sequences analyzed and their sources can be found in Supplementary Tables 1–4.

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

## Acknowledgements

We thank Channa Aluvihare (UMD-IBBR), Rob Harrell (UMD-IBBR), and Megan Fritz (UMD-College Park) for their advice on mosquito rearing and manipulation. We acknowledge the Imaging Core Facility in the department of Cell Biology and Molecular Genetics (UMD-College Park) for access to the Zeiss LSM710 confocal microscope, as well as Amy Beaven for expert training and support on this instrument. We thank members of the Pick lab for feedback on this work and comments on this paper, especially Ebony Argaez for providing excellent mosquito care. This work was supported by the NIH (R01GM113230) to L.P.

## Author contributions

A.M.C.J. and L.P. developed the project. A.M.C.J. and C.S.T. performed experiments. A.M.C.J. and L.P. analyzed the results and wrote the paper.

## Competing interests

The authors declare no competing interests.
