## [Peer Review File · Communications Biology]

Reviewers' comments:

Reviewer #1 (Remarks to the Author):

This is a very well written manuscript that describes how an essential segmentation gene in insects disappeared from the genome of mosquitos by handing off its function to a paralog. The authors should be complemented by the quality of their work and the neat story. I have only very minor comments and look forward to seeing this manuscript in print.

Line 135: "Rather than using an entirely.."

Line 203: Perhaps indicate what test the p value derives from with the same parenthesis. It is a chi-square test for a particular proportion of larvae expected to not have hatched?

Line 342: "For of h"? Also I do not observe a broad anterior stripe in slp in the mosquito embryo as mentioned in the legend.

Lie 345 broad instead of broadly?

Figure 4 (panel c''): It is quite difficult for the non-mosquito expert to understand where the respiratory spiracle is located....can this trait be highlighted or made more visible?

Line 636: Pale blue? The genes appear light grey to me....

A mere suggestion at the authors discretion: why not spend an extra paragraph in the Discussion explaining a potential molecular mechanism for how the novel expression pattern for gooseberry might have come about?

Reviewer #2 (Remarks to the Author):

Holometabolous insects have three Pax group III genes: paired (*prd*), gooseberry (*gsb*) and gooseberry-neuro (*gsb-n*). In *Drosophila*, only *prd* functions as a canonical pair-rule gene during blastoderm patterning, while *gsb* takes over from paired during germband elongation and *gsb-n* is expressed in the central nervous system. In bees, *gsb* expression starts a little earlier, partially overlapping *prd* expression in cellular blastoderms, but also shows some temporal differences before gastrulation. In beetles, *prd* expression is similar to *prd* expression in other species but detailed expression data for *gsb* are apparently not available.

The key findings of Jarvela et al. are: (1) *prd* was lost in the mosquito lineage. (2) In Anopheles (and presumably other mosquitoes), *gsb* takes on the roles of both, *prd* and *gsb*. These two findings are well documented and the quality of in situ and functional data (CRISPR/Cas9 mediated knockdown of *gsb* in Anopheles) are remarkable. The authors conclude from their findings that mosquito *gsb* newly acquired the early pair-rule pattern and thereby allowed the loss of *prd*. The authors call this process "synfunctionalization", following Gitelman (2007) who introduced this term for a hypothetical evolutionary scenario in which a paralog secondarily evolves overlap in gene expression with its sister gene.

Critique

An alternative evolutionary process to account for the data would be that the paralogs had largely overlapping functions (at least with regard to *prd* and *gsb*) prior to the split of mosquitoes from other dipterans and underwent temporal subfunctionalization in the lineage leading to *Drosophila* and independently (apparently to a lesser extent) in the lineage leading to bees. The evidence for rejecting subfunctionalization seems slim and I feel reluctant to accept the author's synfunctionalization hypothesis (which raises many population genetic questions) as the most likely explanation of their findings.

It might be informative to have a closer look at the evolution of the three Pax group III genes. When did these duplications occur and what was the gene's function before duplication? Do the paralogs have shared introns? This would suggest that they evolved by locus duplication and potentially largely overlapping gene expression (as opposed to retrotransposition, which would imply new regulatory cis-sequences for one of the duplicated genes). Most importantly, *gsb* expression and function should be examined in more, and more closely related outgroups to determine whether temporal expression differences between *prd* and *gsb* reported for *Drosophila* and bees evolved independently (to various degrees) or whether they are an ancient heritage of Holometabola, as the authors suggest.

We thank the reviewers for their overall appreciation of our study, especially for noting the high quality of the experimental results, which, in any “new model organism” can be quite challenging. We have addressed all of the critiques and suggestions in our revised version, especially those of reviewer #2, which we believe has improved the manuscript considerably. Responses to individual points are below in blue.

Reviewer #1 (Remarks to the Author):

This is a very well written manuscript that describes how an essential segmentation gene in insects disappeared from the genome of mosquitos by handing off its function to a paralog. The authors should be complemented by the quality of their work and the neat story. I have only very minor comments and look forward to seeing this manuscript in print.

Thank you for this unusually kind review.

Line 135: “Rather than using an entirely..”

We have clarified this phrase: “Rather than employing an entirely distinct segmentation mechanism” (Line 203)

Line 203: Perhaps indicate what test the p value derives from with the same parenthesis. It is a chi-square test for a particular proportion of larvae expected to not have hatched?

Here and later in the same section, we have clarified the test used and moved the value to more clearly indicate that we have compared the control cross embryos to the experimental gsb cross.

Importantly, larvae that fail to hatch and exhibit pair-rule phenotypes represent 25±3% of the eggs collected from gsb crosses, as expected if they are the gsb3xP3GFP/3xP3GFP individuals, which is significantly different from the control crosses where no pair-rule phenotypes were observed (p= 0.0015, Student’s t-test). (Lines 278-282)

“Consistent with straightforward Mendelian genetic expectations, this seven-stripe phenotype was observed in 25±3% of gsb cross progeny but 0% of control cross progeny (p= 0.0114, Student’s t-test).” (Lines 289-291)

Line 342: “For of h”? Also I do not observe a broad anterior stripe in slp in the mosquito embryo as mentioned in the legend.

We have removed the typo “of”, thank you for noticing it! We added arrows to Fig 2e (shown below) to indicate the slp anterior expression that is not part of the pair-rule pattern. (Line 763)

Lie 345 broad instead of broadly?

We have rephrased this sentence: “Several *Anopheles* pair-rule gene orthologs exhibit broad expression at blastoderm” (Line 765-766)

Figure 4 (panel c’): It is quite difficult for the non-mosquito expert to understand where the respiratory spiracle is located....can this trait be highlighted or made more visible?

We have circled the respiratory siphon and the region where it ought to have been in the pair-rule mutant for clarity and added the sentence: “A9 is a respiratory spiracle rather than a typical body segment (circled)” (Line 797)

Line 636: Pale blue? The genes appear light grey to me....

We hope this will be resolved by submission of high quality and high resolution figures to the journal, as opposed to the lower quality pngs embedded in the draft you received. However, we also clarified in the text: “Genes proximal to a *Pax3/7* gene without orthologs on the pictured scaffolds are shown in pale blue with no gene name label” (Line 749). We hope this will aid readers who print or view the paper in greyscale.

A mere suggestion at the authors discretion: why not spend an extra paragraph in the Discussion explaining a potential molecular mechanism for how the novel expression pattern for gooseberry might have come about?

We agree that this is a fascinating topic and hope to revisit it in future work, but for now we are only able to speculate. To this end, we have added “Our analysis suggests that the original gene duplication that created *prd* and *gsb* did not occur by retrotransposon (Sup. Fig. 3). This leaves open the possibility that *cis*-regulatory elements were also duplicated and then modified slightly to create the temporal sequence of *prd* expression followed by *gsb* expression observed in many insect lineages (Fig. 6). In *Drosophila*, this temporal order of expression is ensured by Prd’s direct activation of *gsb*³⁸. Shifting expression dependence to a new activator could simultaneously allow earlier *gsb* expression and reduce the need for Prd, as *gsb* is one of Prd’s few known target genes.” (Lines 343-350)

Reviewer #2 (Remarks to the Author):

Holometabolous insects have three Pax group III genes: paired (*prd*), gooseberry (*gsb*) and gooseberry-neuro (*gsb-n*). In *Drosophila*, only *prd* functions as a canonical pair-rule gene during blastoderm patterning, while *gsb* takes over from paired during germband elongation and *gsb-n* is expressed in the central nervous system. In bees, *gsb* expression starts a little earlier, partially overlapping *prd* expression in cellular blastoderms, but also shows some temporal differences before gastrulation. In beetles, *prd* expression is similar to *prd* expression in other species but detailed expression data for *gsb* are apparently not available.

The key findings of Jarvela et al. are: (1) *prd* was lost in the mosquito lineage. (2) In *Anopheles* (and presumably other mosquitoes), *gsb* takes on the roles of both, *prd* and *gsb*. These two findings are well documented and the quality of in situ and functional data (CRISPR/Cas9 mediated knockdown of *gsb* in *Anopheles*) are remarkable. The authors conclude from their findings that mosquito *gsb* newly acquired the early pair-rule pattern and thereby allowed the loss of *prd*. The authors call this process “synfunctionalization”, following Gitelman (2007) who introduced this term for a hypothetical evolutionary scenario in which a paralog secondarily evolves overlap in gene expression with its sister gene.

Critique

An alternative evolutionary process to account for the data would be that the paralogs had largely overlapping functions (at least with regard to *prd* and *gsb*) prior to the split of mosquitoes from other dipterans and underwent temporal subfunctionalization in the lineage leading to *Drosophila* and independently (apparently to a lesser extent) in the lineage leading to bees. The evidence for rejecting subfunctionalization seems slim and I feel reluctant to accept the author’s synfunctionalization hypothesis (which raises many population genetic questions) as the most likely explanation of their findings.

It might be informative to have a closer look at the evolution of the three Pax group III genes. When did these duplications occur and what was the gene’s function before duplication? Do the paralogs have shared introns? This would suggest that they evolved by locus duplication and potentially largely overlapping gene expression (as opposed to retrotransposition, which would imply new regulatory cis-sequences for one of the duplicated genes). Most importantly, *gsb* expression and function should be examined in more, and more closely related outgroups to determine whether temporal expression differences between *prd* and *gsb* reported for *Drosophila* and bees evolved independently (to various degrees) or whether they are an ancient heritage of Holometabola, as the authors suggest.

We thank the reviewer for commending the quality of our experiments but also for critically analyzing our original hypothesis, forcing us to further probe and re-examine our assumptions and conclusions. After a rather extensive re-analysis of the literature, we find that synfunctionalization remains the most likely, and most strongly supported evolutionary mechanism to explain the loss of *prd*, which occurred specifically in mosquito lineages and well after an initial subfunctionalization of *prd* and *gsb*. We review the evidence and logic for this conclusion below. In addition, we performed new Bayesian and Maximum Likelihood analyses on *Pax3/7* sequences representing as many major arthropod taxa as possible and their outgroup, Onychophora to determine the divergence time of *prd* and *gsb*. We have made a large number of changes to the text and added three additional figures to address these serious concerns. The most relevant new passages are included below.

We have interpreted the large body of literature on pair-rule patterning in insects as supportive of subfunctionalization of *prd* and *gsb* before the emergence of the Holometabola lineage, but the onus is on us to synthesize and present that data to the readers and we failed to do so in our original manuscript. In light of the availability of existing data, we felt that expression experiments in additional species would be redundant. We have added substantial

new text (lines 81-103 and 312-342) and a summary figure to depict that previously published research (see new Figure 6).

Knowing that bees pattern their segments in anterior to posterior order, and comparing their patterns of *prd* and *gsb* expression to that of other species lead us to interpret those results differently than you originally suggested. We see the spatial-temporal differences as supportive of subfunctionalization, even though they are not as temporally distinct as in *Drosophila*, which uses a simultaneous segment patterning mechanism.

Additionally, we performed the necessary bioinformatics analysis to determine when the ancestral *Pax3/7* gene duplicated to generate *prd* and *gsb* (see new Sup. Fig. 2 and text lines 150-157):

We learned that this duplication occurred at the base of Pancrustacea roughly 530 million years ago, or possibly earlier. Together with our original phylogenetic data, we are able to conclude: “Thus, *prd* and *gsb* resulted from an ancient gene duplication event and *prd* was then lost at the base of the mosquito lineage, in spite of its conservation in the majority of insect lineages, including its sister clade within Diptera.” (lines 168-171)

We also assessed exon/intron structure of *prd* and *gsb* in a variety of Pancrustacea lineages and concluded that retrotransposon origin is unlikely to explain the duplication event (Sup. Fig 3, lines 158-163):

Therefore, we note in the text that duplicated *cis*-regulatory sequences with few evolved differences may play a role in this synfunctionalization event: "Our analysis suggests that the original gene duplication that created *prd* and *gsb* did not occur by retrotransposon (Sup. Fig. 3). This leaves open the possibility that *cis*-regulatory elements were also duplicated and then modified slightly to create the temporal sequence of *prd* expression followed by *gsb* expression observed in many insect lineages (Fig. 6)." (lines 343-347)

To address the question of the original *Pax3/7* gene's function, we added some discussion of the literature on expression patterns characterized in long-diverged species: "In long-diverged non-insect arthropods and arthropod outgroups, without clear orthologs to both *prd* and *gsb*, *Pax3/7* genes are expressed during segmentation, usually in a segment polarity-type pattern, but pair-rule-like expression, has also been observed³¹⁻³⁵. The only characterized crustacean *gsb* ortholog is expressed in segment polarity stripes that matches expression in insects lending support to the idea that *prd* and *gsb* subfunctionalized soon after their duplication (Sup. Fig.

2)36. Together these data strongly support that idea the ancestral *Pax3/7* that generated *prd* and *gsb* had a role in segmentation, and suggest that dual roles in pair-rule as well as segment polarity patterning are plausible.” (lines 329-338)

Taken together, the extremely long divergence time of *prd* and *gsb* coupled with existing literature on gene expression and function in Holometabola and recent outgroups strongly support an evolutionary scenario in which the ancestor of Diptera inherited established subfunctionalization of *prd* and *gsb*. The alternative hypothesis, in which *prd* and *gsb* remained redundant for hundreds of millions of years and did not diverge until after mosquitoes and *Drosophila* split is not supported by existing evidence. We don't see enough data supporting this alternative to give it equal discussion in the text. Yet, we acknowledge that duplication of *cis*-regulatory DNA with a lot of conserved overlap in regulation likely aided the syfunctionalization, and also helps to explain why it occurred at other points within Holometabola.

In sum, we thank the reviewer for pushing us to critically re-analyze our assumptions and analyses regarding the evolutionary origin of insect *Pax3/7* genes and timing of *prd* and *gsb* subfunctionalization. As a result, we have added three new figures to the paper and have made major changes in the text in the introduction, results, and discussion sections. We believe that these changes have improved the stringency and quality of our manuscript, and that our findings, which were originally more focused towards developmental biologists, will now be of broader interest to the evolutionary biology community.

REVIEWERS' COMMENTS:

Reviewer #2 (Remarks to the Author):

The authors were able to strengthen their case for an evolutionary scenario, in which *gsb* secondarily adopted an earlier role of its paralog *prd* (synfunctionalization). The study thereby identifies an interesting model system to study this concept.

In their final version, the authors may want to point out that descriptive and functional data from lower dipterans (e.g., *Clogmia*, *Chironomus*) would be suitable to test and potentially strengthen their hypothesis in future studies.

In the abstract, tuning down the last sentences would seem appropriate to me. For example, the authors could state (instead of Thus, ...): "It is proposed, paired was functionally replaced by the related gene, *gooseberry*, in mosquitoes. Our findings suggest a heterochronic shift of *gooseberry* that allowed for redundancy and loss of paired in the lineage leading to mosquitoes." The reason for this suggestion is twofold: a little more caution seems warranted in the absence of data from lower dipterans (other than mosquitoes), and more importantly, the study provides little "mechanistic explanation" of synfunctionalization. However, despite these reservations, it is acknowledged that the authors made an intriguing case for synfunctionalization, thereby providing a study of significant heuristic value.

A minor point: Are any of the introns in conserved positions, or does this question have to remain unanswered due to lack of sufficient sequence conservation?

REVIEWERS' COMMENTS:

Reviewer #2 (Remarks to the Author):

The authors were able to strengthen their case for an evolutionary scenario, in which *gsb* secondarily adopted an earlier role of its paralog *prd* (synfunctionalization). The study thereby identifies an interesting model system to study this concept.

Thank you. We appreciate that you pushed us to clarify and better articulate our logic, which has led to an improved manuscript.

In their final version, the authors may want to point out that descriptive and functional data from lower dipterans (e.g., *Clogmia*, *Chironomus*) would be suitable to test and potentially strengthen their hypothesis in future studies.

We have added to the Discussion: Lines 348-350 “Additional work on *cis*-regulatory elements in mosquitoes as well as functional work in a wider variety of Dipterans would enable us to test these ideas.”

In the abstract, tuning down the last sentences would seem appropriate to me. For example, the authors could state (instead of Thus, ...): “It is proposed, paired was functionally replaced by the related gene, *gooseberry*, in mosquitoes. Our findings suggest a heterochronic shift of *gooseberry* that allowed for redundancy and loss of paired in the lineage leading to mosquitoes.” The reason for this suggestion is twofold: a little more caution seems warranted in the absence of data from lower dipterans (other than mosquitoes), and more importantly, the study provides little “mechanistic explanation” of synfunctionalization. However, despite these reservations, it is acknowledged that the authors made an intriguing case for synfunctionalization, thereby providing a study of significant heuristic value.

With all due respect to the reviewer, we do not feel that our Abstract was overly hyped and, especially given the word limit, prefer not make the proposed changes. Specifically with respect to lower Diptera, we have added text to the Discussion (see above) and, while we agree that this is an interesting point, that information is not an essential component of the story and we have provided extensive phylogenetic data to address this. We also document a spatial change in expression, not merely a temporal shift and do not agree that we can “boil down” the entire story to a heterochronic shift, even though that fits neatly into previous evolutionary hypotheses. For the final point: while our study has not addressed molecular mechanisms underlying evolutionary variation, in the context of evolutionary biology we do consider synfunctionalization to be mechanistic.

A minor point: Are any of the introns in conserved positions, or does this question have to remain unanswered due to lack of sufficient sequence conservation?

In Sup Fig 3, we show that while many members of Pancrustacea have introns in conserved positions, those particular introns were lost by the emergence of Diptera.